Genome-wide identification and characterization of FAD family genes in barley

Cao TingTing 1
Du QingWei 2
Ge RongChao 1 grcgp@sina.com
Li RuiFen 2 liruifen@aliyun.com
1 College of Life Science, Hebei Normal University , Hebei , China
2 Institute of Biotechnology, Beijing Academy of Agriculture and Forestry Sciences , Beijing , China
Mansour Elsayed
Electronic publication date: 2024 Feb 29
Publication date: 2024
Volume: 12
Electronic Location ID: e16812
Received 2023 Sep 1; Accepted 2023 Dec 29
Copyright: © 2024 Cao et al.
Copyright year: 2024
Copyright holder: Cao et al.
License: This is an open access article distributed under the terms of the Creative Commons Attribution License, which permits unrestricted use, distribution, reproduction and adaptation in any medium and for any purpose provided that it is properly attributed. For attribution, the original author(s), title, publication source (PeerJ) and either DOI or URL of the article must be cited.
License URL: https://creativecommons.org/licenses/by/4.0/

Keywords: FAD, Barley, Gene family, Gene expression, Subcellular localization

Funding: Young Scientist Fund of Beijing Academy of Agriculture and Forestry Sciences QNJJ202201 Science and Technology Innovation Project of Beijing Academy of Agriculture and Forestry Sciences KJCX20230117 Collaborative Innovation Center of Beijing Academy of Agricultural and Forestry Sciences KJCX201907-2 This research was supported by the Young Scientist Fund of Beijing Academy of Agriculture and Forestry Sciences (QNJJ202201), the Science and Technology Innovation Project of Beijing Academy of Agriculture and Forestry Sciences (Grant no. KJCX20230117) and the Collaborative Innovation Center of Beijing Academy of Agricultural and Forestry Sciences (Grant no. KJCX201907-2). The funders had no role in study design, data collection and analysis, decision to publish, or preparation of the manuscript.

==============================
Fatty acid desaturases (FADs) play pivotal roles in determining plant stress tolerance. Barley is the most salt-tolerant cereal crop. In this study, we performed genome-wide identification and characterization analysis of the FAD gene family in barley (Hordeum vulgare). A total of 24 HvFADs were identified and divided into four subfamilies based on their amino acid sequence similarity. HvFADs unevenly distributed on six of seven barley chromosomes, and three clusters of HvFADs mainly occurred on the chromosome 2, 3 and 6. Segmental duplication events were found to be a main cause for the HvFAD gene family expansion. The same HvFAD subfamily showed the relatively consistent exon-intron composition and conserved motifs of HvFADs. Cis-element analysis in HvFAD promoters indicated that the expression of HvFADs may be subject to complex regulation, especially stress-responsive elements that may involve in saline-alkaline stress response. Combined transcriptomic data with quantitative experiments, at least five HvFADs highly expressed in roots under salt or alkali treatment, suggesting they may participate in saline or alkaline tolerance in barley. This study provides novel and valuable insights for underlying salt/alkali-tolerant mechanisms in barley.

Introduction

Fatty acids are the major constituent of plant cell membrane. Fatty acid desaturase (FAD) is a critical enzyme in plant lipid metabolism, the FAD gene family mediates the ratio of saturated to unsaturated fatty acid resulting in affecting the plasma membrane fluidity and stability under stress (Zhu et al., 2018). Thus, the FAD gene family plays an essential role in stress response besides plant growth and development. In higher plants, FADs in the same subfamily exhibit remarkably well-conserved amino acid sequences (Chen et al., 2019). FADs contain three conserved histidine motifs (H(X)34H/H(X)23HH/H/Q(X)2 ~ 3HH) Based on their solubility, plant FADs can be classified into two distinct categories: soluble desaturases (FAB2/SAD) and membrane-bound desaturases (Dehghan Nayeri & Yarizade, 2014). The membrane-bound FADs are additionally divided into four distinct subfamilies based on their functions, encompassing FAD4, FAD2/FAD6, FAD3/FAD7/FAD8, and ADS/SLD/DES (Saini & Kumar, 2019). FAB2s typically harbor two conserved histidine motifs (D/EXXH), whereas membrane-bound FADs contain three conserved histidine motifs (H(X)3-4H/H(X)2-3HH/H/Q(X)2-3HH) Notably, the stearoyl ACP desaturase (FAB2/SAD) is the only known soluble FAD in the plastid matrix. All membrane-bound desaturases anchored to the endoplasmic reticulum and plastid membranes. Numerous studies have revealed various plant species contain different FAD family members. For instance, the genome of cucumber (Cucumis sativus) encompasses 23 integral FAD genes (Dong et al., 2016), which is in contrast to the soybean genome (Glycine max) consisting of 29 genes (Yuan et al., 2013), the rice genme (Oryza sativa) containing 20 genes (Chen et al., 2019).

Plant FADs are known to play a crucial role in enhancing the plant’s resilience, including extreme temperatures, drought, high salinity, physical damage, and attacks from pathogens (Saini & Kumar, 2019). In soybean, the increased expression of GmFAD3A noticeably enhanced the plant’s ability to tolerate cold stress (Wang et al., 2019). Similarly, the overexpression of LeFAD3 in tomatoes significantly improved their photosynthetic ability (Wang et al., 2010). It has been reported that the increased expression of AtFAD2 enhances germination and salinity tolerance in Arabidopsis (Zhang et al., 2012). Moreover, the increased expression of TaSSI2 in Arabidopsis significantly strengthened its defense against powdery mildew fungi invasion (Song et al., 2013). Conversely, the overexpression of AtFAD7 has been found to curtail its tolerance to salinity and desiccation (Im et al., 2002). In rice, the upregulated expression of OsFAD7 and OsFAD8 decreased resistance to Phytophthora infestans (Yara et al., 2007). Additionally, the suppression of OsSSI2 significantly improved resistance to rice blast and leaf blight in rice plants (Jiang et al., 2009). Therefore, FAD family members are involved in plant stress tolerance, but the roles of specific FAD members in plant abiotic or biotic stress tolerance need to be investigated in detail.

Land salinization seriously affects the growth and development of crops, ultimately leading to a decline in global agricultural yield (Munns & Tester, 2008). Among the cereal crops, barley can tolerate to middle level of salt. To date there is no report on genome-wide identification and characterization of FAD family genes in barley. In this study, we identified 24 FADs and named them based on their respective chromosomal locations. We conducted a comprehensive analysis on the phylogenetic relationship, gene structure, conserved motifs, chromosomal localization, collinearity, cis-acting elements in the promoters of HvFADs, and their expression under salt and alkali stress treatments. We aim to provide valuable information for identification of key genes underlying salt-tolerant mechanisms in barley.

Materials and Methods

Identification and characterization of FAD family members in the Hordeum vulgare genome

To effectively identify the FAD family members of Hordeum vulgare cv. Morex genome, candidate protein sequences corresponding to the structural domains of FA_desaturatase (PF00487), FA_desatase 2 (PF03405), and TMEM189 (PF10520) were downloaded from the Pfam protein family database (http://Pfam.xfam.org/). Using the SMART database (http://SMART.embl-heidelberg.de/), Markov model (HMM) files were utilized to confirm each putative HvFADs. TBtools software was employed to analyze the physicochemical properties, transmembrane structural domains, and signaling peptides.

Chromosomal distribution of HvFADs

The locations of HvFADs on chromosomes were extracted from the genome annotation files. The genes densities of the whole chromosome were determined and visualized using TBtools software (Chen et al., 2020). Only HvFADs anchored on chromosomes were displayed, for analysis of conserved motifs and domains. Conserved motif analysis of the FAD protein sequences was performed using the classical motif discovery model of MEME-suite 5.3.3 (Bailey et al., 2006). The parameter of the optimum motif was set from 6 to 200, and the maximum number of motifs was set to 10. Conserved domain analysis was performed using TBtools.

Collinearity analysis

To investigate the mechanisms underlying the evolution of HvFADs, MCScanX (Wang et al., 2012) was used to analyze tandem, proximal, and dispersed duplications. Collinearity analysis of HvFAD was conducted locally using the TBtools that map the genes to chromosomal locations (Xie et al., 2018). We identified collinearity resulting from segmental duplication events, as well as tandem duplicates arising from gene duplication events. The MCScanX algorithm inferred that one sequence emerged as a result of the duplication of its counterpart, indicating a gene duplication event.

Phylogenetic analysis

To construct a phylogenetic tree, the protein sequences of T. aestivum L. (68), G. max (30), A. thaliana (27), and O. sativa L (18) were analyzed using the neighbor-joining method. The coding sequences of the full-length genes were aligned using fast Fourier transform (MAFFT) (Kazutaka, Rozewicki & Yamada, 2019). The phylogenetic trees were constructed using MEGA7 software (Arizona State University, Tempe, AZ, USA) with 1,000 bootstrap replicates. The FAD protein sequences of Arabidopsis, Glycine max and rice were downloaded from the Arabidopsis (Philippe et al., 2012), Glycine max (https://www.soybase.org/) and rice (Kawahara et al., 2013) databases for constructing the phylogenetic tree, and the tree was visualized using the iTOL online website (https://itol.embl.de/) (Letunic & Bork, 2019).

Analysis of cis-acting elements in HvFADs promoters

The 1,500 bp sequences upstream the initiation codon (ATG) of each HvFADs were extracted from the Hordeum vulgare genome database using TBtools and subjected to cis-acting elements and transcription factor binding sites prediction analysis using PlantCARE (http://bioinformatics.psb.ugent.be/webtools/plantcare/html/) and PlantTFDB (http://plantregmap.gao-lab.org/binding_site_prediction.php), respectively (Chen et al., 2020).

Protein-protein interaction network

To predict the interactions between HvFAD proteins, FAD protein sequences were submitted to STRING V10 (Szklarczyk et al., 2015). The protein-protein interaction (PPI) networks were then visualized using Cytoscape V3.7.2 (Shannon et al., 2003).

Plant material culture, saline alkali treatment, and quantitative real-time polymerase chain reaction analysis

H. vulgare cv. Morex were cultivated in a light incubator until two leaves and one heart stage. The plants were then exposed to 0, 3, 12 and 24 h of saline alkali treatment. The leaves were collected for RNA extraction. Similarly, roots, stems, leaves and tiller tissues of healthy barley plants were collected at the two leaves and one heart stage for RNA extraction. The quality and concentration of RNA were measured using a NanoDrop 2000 UV spectrophotometer. First-strand cDNA was synthesized using the Novizen Reverse Transcription Kit according to the manufacturer’s protocol. The cDNA was diluted to 100 ng/μl and used as a template for quantitative real-time polymerase chain reaction (qRT-PCR) reaction. Gene-specific primers for RT-PCR and qRT-PCR were designed using NCBI (Table S1). Five genes belonging to four FAD subfamilies were selected to analyze the expression in different tissues, with amplified size ranging from 190 to 230 bp. The PCR program was performed as follows: 94 °C for 2 min; 39 cycles of 95 °C for 5 s, 60 °C for 30 s; 95 °C for 5 s, 65 C for 5 s, and 95 °C for 5 s.

Gene Expression Analysis™ Real-Time PCR Detection System (Bio-Rad, Hercules, CA, USA) and Taq Pro Universal SYBR qPCR Master Mix (Q712-03) Master Kit (Novozymes, Bagsværd, Denmark) were used for gene expression analysis by qRT-PCR experiments using FX96 Touch. Relative fold differences were calculated based on the comparative cycle threshold (2-Δct) using Tublin (F: AGTGTCCTGTCCACCCACTC; R: AGCATGAAGTGGATCCTTGG) as the reference gene. The PCR mixture (20 μl) included 10 μl of 2× Taq Pro Universal SYBR qPCR Master Mix (Novozymes, Bagsværd, Denmark), 10 μm of each primer, 100 ng cDNA template, and nuclease-free water. The PCR program was performed as follows: 94 °C for 2 min; 39 cycles of 95 °C for 5 s, 60 °C for 30 s; 95 °C for 5 s, 65 °C for 5 s, and 95 °C for 5 s.

Transient expression of HvFAD14

Subcellular localization of HvFAD14 in barley protoplasts was analyzed using a transient transformation system as follows. Brifely, a full-length open reading frame (ORF) fragment (1,156 bp) of HvFAD14 was cloned in an empty GFP vector. The resulting construct (HvFAD14-GFP) was then transformed into Fast-T1 chemoreceptor cells. After that, the cells were harvested by shaking the bacteria and centrifugation for bulk plasmid extraction. Protoplasts were extracted from barley yellowing seedling leaves using cellulase and dissociative enzyme solutions, and the reconstructed GFP plasmid was transformed into protoplasts using a 40% PEG solution. After 16 h of dark expression, the expression of HvFAD14 were observed using a Leica fluorescence focusing microscope.

Results

Identification and analysis of HvFADs in barley

We identified a total of 24 HvFADs from barley genome. These genes were named based on their chromosomal locations and phylogenetic relationship (Fig. 1, Table 1), ranging from HvFAD1 to HvFAD24. Systematic analysis of their physical and chemical properties showed that the CDS length of these 24 HvFAD genes ranged from 987 to 1,329 bp, while the lengths of the deduced amino acid sequences ranged from 329 to 469 amino acids. Additionally, the predicted molecular weights (MW) ranged from 37,724.61 Da (HvFAD9) to 52,548.61 Da (HvFAD18), and the theoretical isoelectric points (pI) ranged from 6.04 to 9.37 (Table 1).

Figure 1 FAD genes distribution on barley chromosomes.

The gene density was visualized with the depth of color.

Table 1 Basic information for the Hordeum vulgare FAD gene family members.

Gene_name	Gene_ID	CDS/bp	Number of amino acid/aa	Molecular weight/Da	Theoretical_PI	Instability index	Aliphatic index	Grand average of hydropathicity	
HvFAD1	HORVU.MOREX.r3.2HG0101440.1	1,215	405	44,883.21	7.19	42.7	84.96	−0.241	
HvFAD2	HORVU.MOREX.r3.2HG0101470.1	1,125	375	41,677.74	8.68	41.94	87.33	−0.257	
HvFAD3	HORVU.MOREX.r3.2HG0101500.1	1,215	405	44,883.21	7.19	42.7	84.96	−0.241	
HvFAD4	HORVU.MOREX.r3.2HG0106740.1	1,281	427	47,347	8.47	38.72	85.55	−0.267	
HvFAD5	HORVU.MOREX.r3.2HG0106840.1	1,206	402	44,563	8.4	39.14	85.55	−0.236	
HvFAD6	HORVU.MOREX.r3.2HG0108020.1	1,314	438	48,622.66	8.89	44.32	84.27	−0.157	
HvFAD7	HORVU.MOREX.r3.2HG0129520.1	1,329	443	51,231.58	9.37	51.02	86.75	−0.115	
HvFAD8	HORVU.MOREX.r3.2HG0161410.1	1,176	392	44,559.57	6.04	49.19	73.44	−0.49	
HvFAD9	HORVU.MOREX.r3.2HG0183780.1	987	329	37,724.61	8.79	46.99	86.57	−0.086	
HvFAD10	HORVU.MOREX.r3.3HG0288610.1	1,017	339	38,752.42	9.03	43.19	76.31	−0.297	
HvFAD11	HORVU.MOREX.r3.3HG0307490.1	1,134	378	42,389.33	6.24	41.04	79.29	−0.324	
HvFAD12	HORVU.MOREX.r3.3HG0309010.1	1,251	417	45,352.59	6.87	42.75	78.01	−0.177	
HvFAD13	HORVU.MOREX.r3.3HG0310210.1	1,185	395	44,530.92	6.65	36.32	78.61	−0.409	
HvFAD14	HORVU.MOREX.r3.4HG0354970.1	1,143	381	43,631.91	8.48	44.31	84.93	−0.201	
HvFAD15	HORVU.MOREX.r3.4HG0384980.1	1,308	436	48,836.08	9.2	45.27	86.17	−0.215	
HvFAD16	HORVU.MOREX.r3.5HG0457600.1	1,155	385	43,660.82	5.97	44.8	74.75	−0.381	
HvFAD17	HORVU.MOREX.r3.5HG0466020.1	1,152	384	43,943.35	8.22	43.96	85.03	−0.217	
HvFAD18	HORVU.MOREX.r3.5HG0474290.1	1,407	469	52,548.61	8.45	35.2	90.72	0.135	
HvFAD19	HORVU.MOREX.r3.5HG0486420.1	1,284	428	47,617.36	7.59	46.27	73.93	−0.306	
HvFAD20	HORVU.MOREX.r3.5HG0535350.1	1,173	391	44,667.93	6.05	43.19	80.36	−0.405	
HvFAD21	HORVU.MOREX.r3.6HG0609520.1	1,161	387	44,334.18	8.44	34.6	94.47	−0.02	
HvFAD22	HORVU.MOREX.r3.6HG0609610.1	1,146	382	43,871.95	8.39	30.99	92.09	0.075	
HvFAD23	HORVU.MOREX.r3.6HG0609630.1	1,152	384	44,743.92	8.19	39.75	97.47	0.017	
HvFAD24	HORVU.MOREX.r3.7HG0668240.1	1,236	412	45,390.46	6.87	51.48	72.91	−0.344	
Note:

Vertical columns indicate: Gene name; Gene ID; CDS; Number of amino acid; Molecular weight; Theoretical; Instability index; Aliphatic index; Grand average of hydropathicity.

Phylogenetic relationship analysis of HvFADs

In order to unravel the phylogenetic relationship among HvFADs, we constructed a phylogenetic tree using FADs from Triticum aestivum with 68 members, Glycine max with 30 members, Arabidopsis thaliana with 27 members and Oryza sativa with 18 members. As shown in Fig. 2, we observed that these FADs were categorized into five distinct subfamilies: FAB2, FAD4, FAD3/FAD7/FAD8, FAD2/FAD6, and DES/SLD. The gene number among these FADs of the five species is different from each other, the homology of genes in the same subfamily is high. Among these subfamilies, the HvFAB2 subfamily consists of 14 genes such as FAD1, FAD2, FAD3, FAD4, FAD5, FAD8, FAD10, FAD11, FAD12, FAD13, FAD16, FAD19, FAD20 and FAD24. The HvFAD3/FAD7/FAD8 subfamily comprises FAD6, FAD14, FAD15 and FAD17 four genes. The HvFAD2/FAD6 subfamily includes FAD7, FAD2, FAD22 and FAD23. The DES/SLD subfamily has FAD9 and FAD18 in HvFADs. Furthermore, the FAD4 subfamily does not contain any HvFADs (Fig. 2). So HvFADs have four subfamilies, which is different from other species FAD gene family.

Figure 2 Phylogenetic tree of FAD proteins from T. aestivum L. (Ta) (68), H. vulgare (Hv) (24), G. max (Gm) (30), O. sativa (Os) (18) and A. thaliana (At) (27).

The MEGA 7 neighbor-joining was used to generate the phylogenetic tree. Five subfamilies were categorized and indicated with colors.

Gene structures and conserved motifs of HvFADs

To investigate the distribution of these HvFADs motifs, we identified 10 motifs in barley. Conserved motif analysis using MEME software revealed (Fig. 3A) that HvFAD1, HvFAD3, HvFAD4, HvFAD5, HvFAD24, HvFAD12, HvFAD19, HvFAD11, HvFAD13, HvFAD8, HvFAD16, and HvFAD20 in the FAB2 subfamily have seven identical motifs. Additionally, HvFAD2 and HvFAD1 have six identical motifs. HvFAD10 and HvFAD1 have five identical motifs. HvFAD9, HvFAD18, and HvFAD7 in the DES/SLD subfamily have two identical motifs. Lastly, HvFAD21, HvFAD22, HvFAD23, HvFAD6, HvFAD15, HvFAD14, and HvFAD17 in the FAD2/FAD6 subfamily and FAD3/FAD7/FAD8 subfamily have three identical motifs. In addition, it was found that the conserved motif composition of each subfamily was generally similar. Moreover, the conserved motif composition of FAD3/FAD7/FAD8, FAD2/FAD6 and DES/SLD subfamily is obviously different from that of FAB2 subfamily.

Figure 3 Gene structure and conserved motif analysis of HvFAD genes.

(A) The evolutionary relationship and motif compositions of HvFADs; (B) Intron-exon organizations of HvFADs. Exons were represented by green boxes and introns by black lines. The sizes of exons and introns were estimated using the scale at bottom.

We further analyzed the exon-intron structure of each HvFAD gene (Fig. 3B). The number of introns ranges from 1 to 9, while the number of exons ranges from 1 to 10. Among them, the HvFAD7 has the highest number of introns and exons, with nine introns and 10 exons. Except for HvFAD5, HvFAD10 and HvFAD18, FAB2 and DES/SLD subfamily possess less introns. HvFAD22 and HvFAD23 have no intron. Other members in FAD3/FAD7/FAD8 and FAD2/FAD6 subfamilies have relatively more introns.

Chromosomal location and colinearity analysis of HvFADs

The HvFAD genes randomly and unevenly distributed on all barley chromosomes except chromosome 1 (Chr1H). Chr2H contains the most HvFADs (nine members), followed by 5 HvFADs on Chr5H, 4 HvFADs on Chr3H, 3 HvFADs on Chr6H, 2 HvFADs on Chr4H, and 1 HvFAD gene on Chr7H (Fig. 1). Collinearity analysis was performed to further investigate gene duplication events in the HvFADs family. Our data revealed a single pair of segmentally duplicated genes (Fig. 4). Segmental duplication events may be a main cause for the HvFAD gene family expansion.

Figure 4 Collinearity among HvFADs.

The members of the barley HvFADs and respective chromosomal locations were labelled. The scale on each chromosome is in megabases (Mb). The segmental duplication genes are connected by straight red line.

Analysis of cis-acting elements in HvFADs promoters

A comprehensive investigation was conducted to clarify the cis-regulatory elements found in the promoter domains of HvFADs. The identified cis-acting components within these promoters were classified into four main groups: light-responsive elements, hormone-responsive elements, stress-responsive elements, and elements essential for plant growth and maturation (Fig. 5). Light-responsive elements were present in all HvFAD promoters, with the Sp1 element being the most prominent with 148 occurrences. However, it is important to note that the promoters of HvFAD11, HvFAD23, and HvFAD24 did not contain this particular element.

Figure 5 The distribution of cis-acting elements in promoters of HvFADs.

Each cis-acting element and function was showed on the right, and the corresponding number of them was indicated by the color scale.

For hormone-responsive elements, the methyl jasmonate (MeJA) response element has emerged as the predominant one with 118 instances, closely followed by the abscisic acid (ABA) response element with 113 instances. All HvFAD promoters, except HvFAD8, contain ABA response motifs (specifically ABRE and DRE1). The cis-regulatory motifs associated with the MeJA response, including the TGACG and CGTCA patterns, are present in the promoters of 20 HvFADs. HvFAD4, HvFAD6, HvFAD13 and HvFAD12 possess promoters that feature growth hormone, ethylene, SA and gibberellin response motifs, respectively. Moreover, all HvFAD promoters have at least two hormone-responsive motifs. Notably, HvFAD4 and HvFAD5 promoters were characterized by the inclusion of all the previously mentioned hormone-responsive motifs.

In regard to stress-responsive motifs, the drought-responsive elements, specifically MYC, as-1 and MBS were found to be present with a total count of 167. However, the MYC element was not found in the promoters of HvFAD11 and HvFAD16, but was present in the remaining 22 HvFAD promoters. The as-1 motif had a wide distribution, only being absent from the promoters of HvFAD9, HvFAD13, HvFAD17 and HvFAD24. The MBS elements were found in a more limited number, featured in only 16 HvFAD promoters. Additionally, defense- and stress-responsive motifs, including STRE, CARE and TCrich repeat sequences, were highly prevalent, totaling 77. STRE patterns were found in 18 promoters, CARE in a single promoter, and TCrich repeat sequences in six promoters. Elements related to drought, trauma, pathogens (WRE3, box, and WUN motifs), defense and stress, temperature fluctuations, dehydration (DRE core), anaerobic induction (ARE) and hypoxia-specific induction (GC motif) were distributed among 24, 21, 19, 11, 7, 16 and 9 HvFADs, respectively, as shown in Fig. 5.

Analysis of protein-protein interaction network of HvFADs

As shown in Fig. 6, members of HvFADs interact with each other. The highest efficiency of interaction was found between HvFAD7 and HvFAD17. The possible interactions among HvFADs may provide the information for the research on their biological functions (Fig. 6).

Figure 6 The protein-protein interaction network for HvFADs based on their orthologs in Arabidopsis.

HvFAD expression under different stresses by qRT-PCR

HvFADs have been reported to function under various abiotic and biotic stresses (Saini & Kumar, 2019). In this study, qRT-PCR was utilized to detect changes in the expression of HvFADs under saline and alkaline stress conditions. According to the transcriptomic data of barley treated with salt and alkali, six genes with higher expression levels and significant changes in expression levels under alkali and salt treatment were selected for qRT-PCR analysis: three genes from the FAB2 subfamily (FAD8, FAD11, FAD13), two genes from the FAD3/FAD7/FAD8 subfamily (FAD14, FAD15), and one gene from the FAD2/FAD6 subfamily (FAD21). As shown in Fig. 7, the expression levels of HvFAD8, HvFAD11, HvFAD13, and HvFAD21 were significantly up-regulated under salt treatment, reaching between 1.3 and seven times, respectively. Among them, the expression of HvFAD11 changed the most, up to seven times. In addition, the expression levels of HvFAD14 and HvFAD15 were significantly up-regulated at 3 h of salt treatment, and the expression level of HvFAD15 was up-regulated to about 65 times at 3 h of salt treatment, but gradually decreased with the increase of salt treatment time. The results showed that these genes responded to salt stress (Fig. 7).

Figure 7 Expression analysis of HvFADs in barley roots under NaCl treatment.

Relative expression of HvFADs genes in barley following the treatments of 200 mm NaCl for 0, 3, 12 and 24 h was validated through the qRT-PCR method. Bars represent the mean values of three replicates ± standard deviation (SD). Values represent average and standard deviation of three biological replicates and ns denotes not Statistically, **** denotes p < 0.0001, ** denotes p < 0.01, * denotes p < 0.05.

As shown in Fig. 8, the expression levels of HvFAD13 and HvFAD14 were significantly up-regulated with the increase of alkali treatment, reaching about nine to 14 times, respectively. The expression levels of HvFAD8 and HvFAD21 had no significant change after alkali treatment for 3 h, but were significantly down-regulated after alkali treatment for 12 h. The expression levels of HvFAD21 were significantly up-regulated after alkali treatment for 24 h. The expression of HvFAD11 was significantly down-regulated to about 0.5 times under alkaline treatment. The expression of HvFAD15 was significantly up-regulated at 3 h with alkali treatment, reaching about 50 times, but significantly down-regulated at 12 and 24 h with alkali treatment. Finally, after 24 h alkali treatment, the expression levels of HvFAD13, HvFAD14 and HvFAD21 were significantly up-regulated, while the expression levels of HvFAD11 were significantly down-regulated. The results showed that these genes also responded to alkali stress (Fig. 8).

Figure 8 Expression analysis of HvFADs under in barley roots NaHCO3 treatment.

Relative expression of HvFADs genes in barley following the treatments of 200 mm NaHCO3 for 0, 3, 12 and 24 h was validated through the qRT-PCR method. Bars represent the mean values of three replicates ± standard deviation (SD). Values represent average and standard deviation of three biological replicates and ns denotes not Statistically, **** denotes p < 0.0001, *** denotes p < 0.001, ** denotes p < 0.01, * denotes p < 0.05.

Localization analysis of HvFAD14 protein

In order to investigate the subcellular location of HvFADs, HvFAD14 with high expression in barley FAD gene family was cloned. The localization of HvFAD14 protein was distributed in the endoplasmic reticulum compared to the control (Fig. 9).

Figure 9 Subcellular localization of HvFAD14 in barley protoplast.

The scale represented 5 µm. GFP was used as control showing green fluorescence protein signal.

Discussion

To date, the FAD gene family has emerged as a subject of considerable interest. It has been recognized and characterized across an extensive array of plant species, with each species exhibiting a distinct number of family members (Paterson et al., 2009). In our study, we focused on the barley (H. vulgare) genome and identified 24 distinct FADs. These HvFADs can be classified into four notable subfamilies. Interestingly, we did not find any members of FAD4 subfamily and ADS members in barley. This absence of ADS members is consistent with previous research on the banana FAD gene family (Cheng et al., 2022). However, dicotyledonous plants such as soybean and Arabidopsis do contain ADS members, suggesting that the ADS subfamily emerged after the divergence of monocotyledons and dicotyledons (Zhang et al., 2021). All members of the HvFAB2 subfamily possess more similar motifs compared with those of the other three subfamilies. Notably, the protein sequences of the SLD/DES, FAD2/FAD6, and FAD3/FAD7/FAD8 subfamilies are highly similar, indicating a closer relationship between these subfamilies compared to the FAB2 subfamily. Moreover, HvFAD members within the same subfamily exhibit similar intron/exon structures and intron phases. Similar results were also observed in wheat (Hajiahmadi et al., 2020) and rice (Chen et al., 2019), demonstrating a high conservation of the FADs gene family. The study of promoter regions is crucial for understanding gene interactions and functions. Transcription factors play a crucial role in coordinating signaling cascades alongside abiotic stresses (Hao et al., 2021). Functionally, these molecular regulators bind to the promoter regions of target genes, either activating or repressing target genes (Lindemose et al., 2013). The presence of TGACG and CGTCA motifs in genes is associated with a response to methyl jasmonate (Dong & Shang, 2013). In a detailed exploration of regulatory elements, the ABRE and MBS motifs have been identified as key modulators of responses to abscisic acid (ABA) and drought conditions, respectively. Jasmonates, on the other hand, are fundamental in various physiological processes, including seed germination, cellular senescence, and responses to both biotic and abiotic stresses. The ABRE motif, characterized by the TACGGTC sequence, becomes enhanced in the presence of ABA. These cis-elements in HvFADs may involve in respond to salinity or alkali stress.

Gene duplication is an essential process in the evolution of various organisms, enabling the development of new structures and functions. Whole genome doubling (WGD) is a significant evolutionary event that occurs in plants, animals, and fungi. It leads to the simultaneous generation of numerous duplicated genes. Upon analyzing covariance, it was discovered that a pair of genes had undergone segmental duplication. Of the 24 desaturase genes, eight of them were involved in repeated events. The tandemly duplicated genes were located on chromosomes chromosome 2H (Chr2H) and chromosome 6H (Chr6H). These findings indicate that gene duplication events play a role in the expansion of the barley desaturase family.

In regard to specific gene promoters, it has been found that the oil palm’s EgFAD8 promoter contains a wide range of stress-responsive and phototropic elements (Cao et al., 2016). This particular gene shows a strong preference for low-temperature and reduced light conditions. Similarly, the transcriptomic analysis of the Kale-type oilseed rape BnFAD2-C5 has revealed an upregulation of gene expression in response to SA and JA stimuli. Interestingly, distinct SA-responsive and JA-responsive elements have been identified within specific regions of its promoter (Liu et al., 2016). Moreover, both the BnFAD2A5-1 and Sesame SeFAD2 promoters incorporate ABA-responsive cis-elements (ABRE), with their gene expression being induced by ABA (Xiao et al., 2014). Our investigation has shown a significant presence of MeJA, ABA and SA stress-related elements within the HvFAD promotes. It is worth noting that these elements are involved in light, hormonal, stress, and developmental responses, suggesting a potential broad-spectrum expression of HvFADs in response to various hormones and abiotic stresses. However, it is important to acknowledge that the presence of a cis-acting element does not always guarantee gene expression under corresponding stresses or hormonal cues. This paradox can be attributed to the complex mechanisms governing gene expression and the limitations of computational tools in accurately predicting promoter cis-elements (Liu et al., 2014). Therefore, empirical methods like qRT-PCR remain essential for accurately identifying functional regulatory elements within HvFADs promoters.

An analysis of tissue expression data of HvFADs showed that the homologous genes exhibited similar expression patterns, indicating their functional conservation (Le et al., 2012). qPCR results indicated that HvFADs were involved in the regulation of abiotic stress in barley, specifically under salt stress and alkali stress. HvFAD14, HvFAD15 and HvFAD21 play crucial roles in controlling salt stress and alkali stress in barley. This result is closely linked to their genetic structure. Tissue-specific expression profiles help identify specific genes and their roles during specific developmental stages. FAD is a key enzyme regulating the biosynthesis of polyunsaturated fatty acids. The results of protein localization analysis showed that FAD is located in the endoplasmic reticulum and plays an important role in the synthesis pathway of polyunsaturated fatty acids.

Conclusions

In this study, we identified 24 FADs from the barley genome using bioinformatics methods. Phylogenetic analysis has revealed that HvFADs can be categorized into four subfamilies (FAB2, ADS/SLD/DES, FAD2/FAD6 and FAD3/FAD7/FAD8). HvFADs unevenly distributed on six of seven barley chromosomes. Segmental duplication events may be a main cause for the HvFAD gene family expansion. The same HvFAD subfamily showed the relatively consistent exon-intron composition and conserved motifs of HvFADs. Cis-element analysis in HvFAD promoters indicated that the expression of HvFADs may be subject to complex regulation, especially stress-responsive elements. Quantitative results showed that at least five HvFADs highly expressed in roots under salt or alkali treatment, suggesting they may participate in saline or alkaline tolerance in barley.

Supplemental Information

Supplemental Information 1 The sequences of FAD proteins in barley.

All the pictures in the article are analyzed using the protein sequences of 24 FAT members screened from barley.

Supplemental Information 2 The specific primers for qRT-PCR.

Each gene corresponds to a pair of primers for qRT-PCR

Supplemental Information 3 The specific primers for subcellular localization.

HvFAD14 homologous recombinant primer for subcellular localization.

Supplemental Information 4 qRT-PCR raw data.

Supplemental Information 5 Expression analysis of HvFADs transcriptome data.

Supplemental Information 6 MIQE Checklist.

Additional Information and Declarations

Competing Interests

Author Contributions

Data Availability

The authors declare that they have no competing interests.

TingTing Cao conceived and designed the experiments, performed the experiments, analyzed the data, prepared figures and/or tables, and approved the final draft.

QingWei Du analyzed the data, prepared figures and/or tables, and approved the final draft.

RongChao Ge conceived and designed the experiments, authored or reviewed drafts of the article, and approved the final draft.

RuiFen Li conceived and designed the experiments, authored or reviewed drafts of the article, and approved the final draft.

The following information was supplied regarding data availability:

The barley FAD protein, qRT-PCR and subcellular localization primer are available in the Supplemental Files.

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
