# Peer review of "Genome-wide identification and characterization of FAD family genes in barley"

_PeerJ, doi:10.7717/peerj.16812_

## Round 0.1 · original submission · Major Revisions

Dear Authors,
Thank you for your submission "Genome-wide identification of the FAD gene family in barley and gene expression under saline stress " to PeerJ.

Two reviewers have reviewed the manuscript and based on their evaluation it requires Major Revisions.

Please address the changes recommended by reviewers and resubmit the manuscript.

Best regards,
Elsayed Mansour

**Language Note:** The review process has identified that the English language must be improved. PeerJ can provide language editing services - please contact us at copyediting@peerj.com for pricing (be sure to provide your manuscript number and title). Alternatively, you should make your own arrangements to improve the language quality and provide details in your response letter. – PeerJ Staff

·

Basic reporting

The manuscript is well written but still needs some minor changes

Experimental design

The study demonstrates a good experimental design.

Validity of the findings

No comments

Additional comments

I reviewed the paper titled "Genome-wide identification of the FAD gene family in barley and gene expression under saline stress". The authors selected several members of the HvFADs gene family to undergo quantitative experiments on barley that was subjected to salt and alkali stress treatments. Based on the expression results, they analyzed genes that exhibited significant expression changes under salt and alkali stress for their subcellular localization.
-Comments and Suggestions for Authors
Abstract
The abstract is well written.
Introduction
-The introduction section is comprehensive and well written.
-Please find more corrections as track changes in the manuscript pdf file.
Materials and methods
- In the expression analysis by qRT-PCR, if possible, I suggest the authors to mention the size of the amplicons and from either the 5' end or the 3' end they were amplified.
-The authors should add more details about the method of transient transformation.
- In line 182, the highlighted sentence needs to be revised as the follows "Subcellular localization of HvFAD14 in barley protoplasts was analyzed using a transient transformation system."
- In line 183, please rewrite as the follows "A full-length open reading frame (ORF) fragment (1156 bp) of HvFAD14 was cloned in an empty GFP vector. The resulted construction (HvFAD14-GFP) was transformed into Fast-T1 chemoreceptor cells"
- In line 189, the expression of HvFAD14 was observed using.....not the plasmid
-Please find more corrections as track changes in the manuscript pdf file.
Results
-The results section is well written.
- In line 198, rewrite this sentence to "The lengths of the deduced amino acid sequences of these genes ranged from 329 to 469 amino acids."
- Please find more corrections as track changes in the manuscript pdf file.
Discussion
-The discussion section is well written.
- The result of localization analysis of HvFAD14 protein needs to be well discussed in discussion section.
The study of protein subcellular localization is important to elucidate protein function.
Conclusion
-The conclusion section is well written.
References
-Please unify the style according to the journal instructions
-All scientific names both in the text and in the References should be Italics.
Figure 2: Please add more information such as "Bootstrap values were calculated from 1000 replications and only the values with ?????? % bootstrapping were considered significant, and are indicated on the branch nodes. "

Reviewer 2 ·

Basic reporting

1) The study did not describe the methodology before "Analysis of cis-acting elements in HvFADs promoters", please supplement the research method. Please also supplement the sources of soybean FAD data in the "Phylogenetic analysis", as well as the variety and cultivation environment of the materials in the "Plant materials".
2) There is a lot of repetition in lines 202 and 240 regarding chromosome localization, so please streamline the article.
3) There is a discrepancy between the research results and the chart data in this article, as well as inconsistencies between the previous and subsequent text (lines 304, 308, and Figure 8), which must be rechecked.
4) The study contains several instances where the figure notes do not match the content of the images (e.g., Figure 1, Figure 3, Figure 5, etc.), so please double-check.
5) The description of gene plural is incorrect. Please select either "HvFADs" or "HvFAD gene" and carefully review the entire content.
6) Please review the full text carefully, as there are many low-level errors in this paper. For example: repetition of content (lines 307 and 308); writing errors (line 231) and inconsistencies (lines 282 and 284); irregularities in the citation of literature (lines 419 and 435); and irregularities in the use of punctuation and capitalisation.
7) The language expression ability needs to be improved, and there are ambiguity and grammar errors in expression (e.g., lines. 154, 182, 261, and 274). It is strongly recommended that you seek help from colleagues who are fluent English speakers.
8) Finally, as there are too many problems, it is recommended that the article be carefully checked and corrected from beginning to end.

Experimental design

no comment

Validity of the findings

no comment

---

## Round 0.2 · Minor Revisions

The article presents valuable insights into the stress response of the FAD gene family; however, it still requires significant revisions to address several issues highlighted by Reviewer 2. The manuscript needs improvement in terms of clarity, focus, grammar, and formatting. With substantial revisions addressing the reviewers' comments, the article could be reconsidered for publication. Adhering to these revisions should enhance the clarity, accuracy, and overall quality of the manuscript, making it more suitable for publication.

·

Basic reporting

no comment

Experimental design

no comment

Validity of the findings

no comment

Additional comments

The authors have made the changes I suggested in the last review. I recommend its publication in this journal.

Reviewer 2 ·

Basic reporting

1.The title of the article is relatively single-faceted, only involving one type of stress. Besides, the focus of article is not prominent. There is less support for the article on the salt-alkali stress of the FAD gene.
2.The part of “Introduction, Result, and Discussion” pay attention to tense issues. The description of “Results” section is cumbersome, failing to highlight the key content of the expression, and resulting in a single way of expression.
3. In line 62-65, there is not much difference in the number of genes and no sharp contrast. In addition, the sentence structure should be simplified.
4.The article lacks information on primers for detecting expression level genes, segmental replication, and tandem replication.
5.The “Phylogenetic relationship analysis of HvFADs” section does not specify the number of gene family members for each species. And the explanation of the evolutionary tree is not clear enough.
6.Grammar problems are still serious, such as singular and plural problems(line 132), and capitalization issues with subheadings(line 151).
7.Why does the eggplant species appear in the annotations of Figures 1 and 2? Theoretical PI missing word in Table 1. And in line118, without indicating the unit for setting conditions.
8.The iTOL online does not provide links(line137) or literature support, and there is ambiguity in the description on line 362. Please revise them.
9.Why only five gene family members were selected for stress treatment? What is the basis? At least every subfamily should be involved.
10.There are many detailed issues in the article. Please revise the entire text one by one, seek help from experienced people, and learn to refer to the expressions in published articles. The format issue is serious, and line130 has Chinese format; 2-ΔΔCT expression error, repeated use of punctuation marks; chromosome 2H (Chr2H) format is inconsistent, and the chart sizes in Figures 7 and 8 are inconsistent.
11.There are still many issues with italicized formatting in the article, such as in cis-acting regulatory elements, “cis” should be italicized; Oryza sativa L (Line 65) , Phytophthora infestans italicized(line 80,) etc., including but not limited to chart titles, which should be checked throughout the text.

Experimental design

.

Validity of the findings

。。

Additional comments

---

## Round 0.3 · accepted · Accept

The authors have addressed all the reviewers' comments, rendering this manuscript ready for publication.